# Rescue and quality control of historical geomagnetic measurement at SheShan Observatory, China

Suqin Zhang[1], Changhua Fu[1], Jianjun Wang[2], Guohao Zhu[3], Chuanhua Chen[4], Shaopeng He[5], Pengkun Guo[5] and Guoping Chang[5]

[1]Institute of Geophysics, China Earthquake Administration, Beijing, 100081, China
[2]Earthquake Administration of Gansu Province, Lanzhou, 730000, China
[3]Shanghai Earthquake Agency, Shanghai, 200062, China
[4]Earthquake Administration of Shangdong Province, Jinan, 250014, China
[5]Hebei Earthquake Agency, Hebei Province, Shijiazhuang, 050022, China

*Correspondence to*: *Changhua Fu (*pangzhayu@139.com)

**Abstract.** The Sheshan geomagnetic observatory (IAGA code SSH), China was built in Xujiahui, Shanghai in 1874 and moved to Sheshan, Shanghai at the end of 1932. So far, SSH has a history of nearly 150 years. It is one of the earliest geomagnetic observatories in China and one of the geomagnetic observatories with the longest history in the world. In this paper, we present the rescue and quality control of the historical data at SSH from 1933 to 2019. The rescued data are the absolute hourly mean values (AHMVs) of D, H and Z components. Some of these data are paper-based records, and some are stored in a floppy disk in BAS, DBF, MDB and other file storage formats. After digitization and format transformation, we imported the data into the Toad database to achieve the unified data management. We performed statistics of completeness, visual analysis, outliers detects and data correction on the stored data. Then we conducted the consistency test of daily variation and secular variation (SV) by comparing the corrected data with the data of the reference observatory, and the computational data of the COV-OBS model, respectively. The consistency test reveals good agreement. However, the individual data should be used with caution because these data are suspicious values, but there is not any explanation or change registered in the available metadata and logbooks. Finally, we present examples of the datasets in discriminating geomagnetic jerks and study of storms. The digitized and quality controlled AHMVs data are available at: https://doi.org/10.5281/zenodo.7005471 (zhang et al, 2022).

## 1 Introduction

Geomagnetic observation data contains abundant solar-terrestrial spatial information, which is widely used in geoscience and space science research. The observation data with time resolution of one second to one hour are usually used to study various short-period magnetic event such as pulsation, geomagnetic crochet, geomagnetic bay and magnetic storm (Zhao et al., 2019), and to monitor and predict the electromagnetic environment in solar-terrestrial space. At the same time, it also has important applications in detecting underground electrical structures and evaluating the impact of geomagnetic induced current (GIC) on underground metal pipe network, transmission network, communication cables, high-speed railway lines

and other major projects (Kappenman, 1996; Bolduc et al., 1998, 2002; Boteler et al., 1998; Liu et al., 2008, 2016; Liu et al., 2009; Guo et al., 2015). Observation data with time resolution of 1h to hundreds of years are usually used for the study of geomagnetic field and its secular variation, such as geomagnetic jerk (Courtillot and Mouël, 1984; Xu, 2009), magnetic pole movement, dipole magnetic moment change, westward drift, etc., which are of great significance for understanding the material flow inside the core and at the core mantle boundary.

The development and application of geomagnetism depends on long-term data accumulation. The long-term operation of geomagnetic observatory is very important for the study of the geomagnetic field (Linthe et al., 2013). It is especially valuable to study the variation characteristics of the geomagnetic field from decades to hundreds of years (Clarke, 2009; Zhang et al., 2008b). Using the latest scientific and technological means to analyse the geomagnetic continuous observation data as long as possible, to obtain the variation information of geomagnetic field, has always been a method often used by scientific researchers. However, not all data can be directly provided to researchers, because some data still exist only in the form of hard copy, and even some data face the risk of serious damage and loss due to improper storage conditions. Therefore, it is very important to rescue and digitize these data as soon as possible. High quality data are the basis of scientific research and the prerequisite for obtaining valuable results (Linthe et al., 2013). Scientists around the world have paid more and more attention to the accumulation of observation data, the rescue of historical data and the sharing of scientific data resources (Curto and Marsal, 2007; Peng et al., 2007; Korte et al., 2009; Dawson et al., 2009; Morozova et al., 2014, 2020; Sergeyeva et al., 2020; Dong et al., 2009; Zhao et al., 2017; Thomson, 2020).

The rescue, recovery, digitization and the quality control of historical geomagnetic data are of extraordinary importance for the geomagnetic community (Rasson, et al., 2011). This paper presents the collection, collation, digitization, the quality control and the correction of the historical data of Sheshan Geomagnetic Observatory (International Association of Geomagnetism and Aeronomy code SSH) from 1933 to 2019. SSH Geomagnetic Observatory is the geomagnetic observatory with the longest history in China. Although many efforts have been made (Gao et al.,1993), the existing data are still insufficient. Our work aims at filling the lack of observation data at SSH observatory since 1933, presenting the absolute hourly mean values (AHMVs) data collected from 1933 to 2019.

This paper is organized as follows. Section 2 describes the data acquisition method, providing information about SSH observatory history, data sources and the digitization method. Section 3 introduces the quality control of the digitized data. Section 4 describes the correction of the selected problem data. Section 5 describes the validation of the corrected series by comparing with reference series and section 6 presents application examples of the datasets. Concluding remarks are given in section 7.

## 2 Data Production Methods

### 2.1 The Sheshan Geomagnetic Observatory

The first step of the data rescue process was to collect resources scattered in different locations, which exist in various forms, including data and metadata that may have an influence on data rescue (observatory relocation, instrument replacement, replacement of observers, environmental change, etc.). We have carefully examined the documentation stored in the SSH, Geomagnetic Network of China (GNC) and reference room of Institute of Geophysics, China Earthquake Administration (IGP, CEA). The reference room is a resource centre of IGP, CEA, used to collect books, journals, papers, monographs, Unpublished reports s, internal textbooks, research reports, reference documents and scientific research achievements related to the discipline. It took us nearly two months to collect resources. The documentation consulted includes *Observatory Communication Journal*, *Geomagnetic Observation Report*, *Chronicles of China Geomagnetic Observatory* and postal letters. The metadata is mainly stored in the *Chronicles of China Geomagnetic Observatory and Geomagnetic Observation Report*. An example of the cover of the bibliographic documents is shown in Fig. 1. The data of 1933-1954 were recorded in the *Geomagnetic Observation Report*. The observation was interrupted from April 1945 to December 1946 due to war. The data of 1955-1994 were stored in the DBF format. The data of 1995-2001 were stored in the BAS format. The data from 2002 to 2006 were stored in the Access database in the MDB format. DBF, BAS and MDB are all data file storage formats. The DBF is a tabular data file stored in binary and is the database format used by dBase and FoxPro databases in DOS systems. The BAS file format is written in the BASIC language, a plain-text data storage format. The MDB format is a storage format used by Microsoft Access software that can generally be opened directly with ACCESS. The data of 2007, 2008 and 2010 were lost for unknown reasons. The data from August to December 2011, and July to October 2019 were missed due to the failure of absolute observation instruments. Data for other years from 2009 to 2019 are stored in Oracle database.

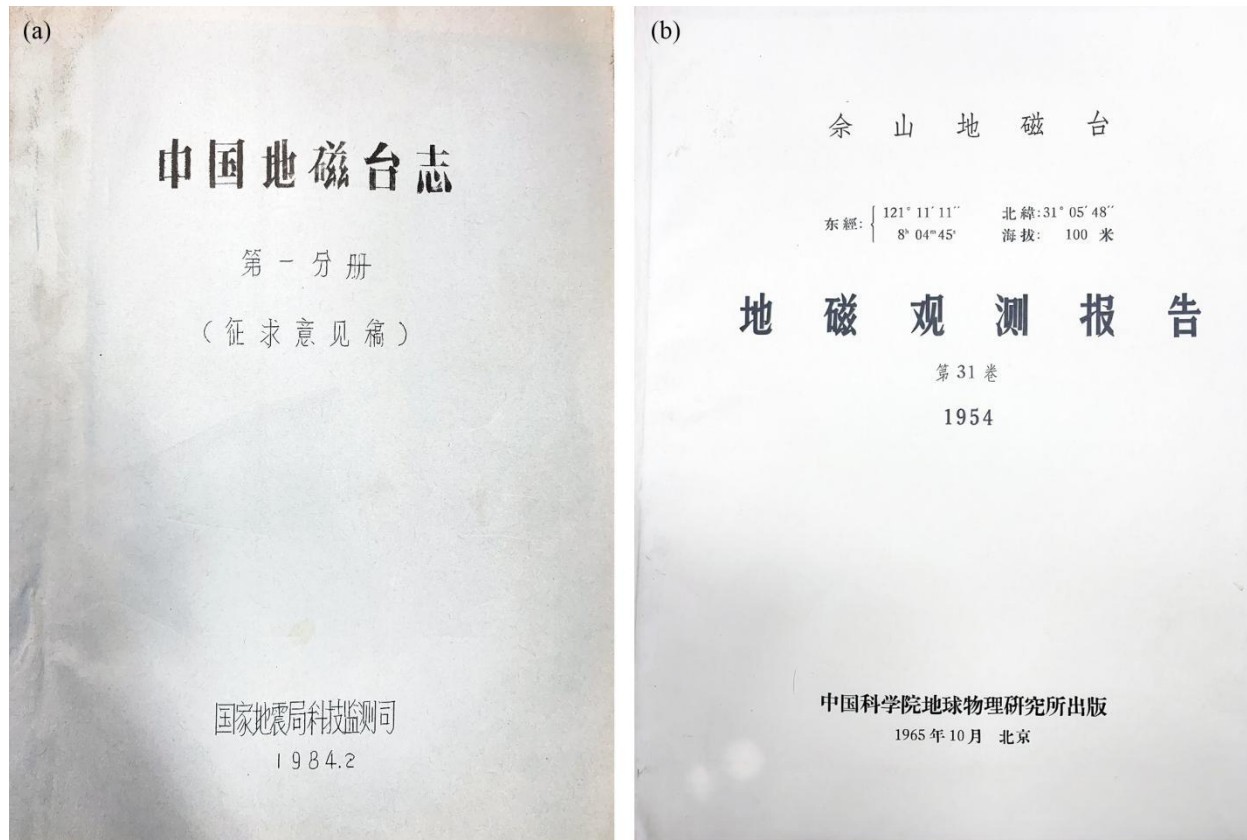

**Figure 1 (a) Cover of the *Chronicles of China Geomagnetic Observatory* (Department of science, technology and monit*oring*, CEA, 1984) and (b) *Geomagnetic Observation Report* (Institute of Geophysics, Chinese Academy of Sciences, 1965)**

The Sheshan Geomagnetic Observatory (SSH) is presently run by IGP, CEA, and has been in operation for almost 150years. Its predecessor is Shanghai Xujiahui Observatory and Jiangsu Kunshan Lujiabang observatory. It was established in Xujiahui in 1874 and began continuous geomagnetic observation since then. It was moved from Xujiahui to Lujiabang in 1908 and was moved from Lujiabang to Sheshan in December 1932. The Sheshan Geomagnetic Observatory is located in Sheshan (Latitude: 31.1° N, Longitude:121.2°E), 20 km to the southwest of Shanghai city. The geology of the vicinity of the observatory is Upper Jurassic to Down Cretaceous Andesite. The gradient of the field is about 2-3 nT/m. The earliest absolute house and recording room were built in 1932, they are made of non-magnetic material. The regular observation began in 1933.

Table 1 shows the absolute and relative instruments in SSH observatory from 1933 to 2019 and the measured geomagnetic elements at different periods. The information was retrieved from the bibliographic documents mentioned above. The first instrument set included as absolute instruments an Elliott (D measurement), a Smith (H measurement) and a Schulze (I measurement) since 1933. The continuous recordings of magnetic variations of D, H and Z were obtained respectively with a

horizontal variometer (Toepfer) and a vertical intensity variometer (Godhavn) since 1933. Later, a few replacements of instruments took place in SSH observatory (Table 1). During this period, many jumps were seen in the relative recorded data due to the adjustment of the variometer, the lightning stroke, earthquake and other reasons. These jumps have been corrected by the baseline, so that the absolute value is not affected. By the 2000, the SSH observatory was equipped with digital instruments. On Jan.1, 2003, the Schmidt Standard Theodolite was replaced by DIM-100/353766 Fluxgate Theodolite and an Overhauser Effect Proton Precession Magnetometer GSM-19F replaced Proton Precession Magnetometer CZM-2.

**Table 1 Summary of instruments in the period from 1933-2019 at SSH**

| Component | Absolute measurements | | Relative measurements | |
| | Date | Instrument name and type | Date | Instrument name and type |
|---|---|---|---|---|
| D | 1933-1969.6 | Magnetometer (Elliott/49) | 1933-2000 | horizontal variometer (Toepfer) |
| | 1969.6-2002 | Standard Theodolite(Schmidt /572144) | 2000-2019 | Fluxgate magnetometer (FGE) |
| | 2003-2009 | Fluxgate Theodolite (DIM-100/353766) | | |
| | 2009-2019 | Fluxgate Theodolite (MINGEO DIM) | | |
| H | 1933-1992 | Magnetometer (Smith/35416) | 1933-2000 | horizontal variometer (Toepfer) |
| | | | 2000-2019 | Fluxgate magnetometer (FGE) |
| I | 1933-1992 | Geomagnetic induction instrument (Schulze/42) | | |
| | 1993-2009 | Fluxgate Theodolite (DIM-100/353766) | | |
| | 2009-2019 | Fluxgate Theodolite (MINGEO DIM) | | |
| Z | | | 1933-2000 | vertical intensity variometer (Godhavn) |
| | | | 2000-2019 | Fluxgate magnetometer (FGE) |
| F | 1981-1985 | Proton Precession Magnetometer (CHD-5/10) | | |
| | 1985-2002 | Proton Precession | | |

| | Magnetometer (CZM-2) |
|---|---|
| | Overhauser Effect Proton |
| 2003-2019 | Precession Magnetometer |
| | (GSM-19F) |

## 2.2 Data digitization

Because some records are handwritten or manual mimeographed, it is impossible to automate the digitization process. To
facilitate the digitization and further application of these records, all the documents were photographed. It is also useful for
checking the consistency of digitized data and source data in the future. For old paper copies it is not good to be carried
around too much and as soon as we have a digital picture, which is fast to make, we can bring the respective paper again to
its normal archive place with the usual temperature, humidity etc. Using the character recognition program to recognize the
photos and compare the consistency with the paper data, it was found that the recognition effect of character was not ideal. It
may be due to the light color of the handwriting, or some of the handwriting is fuzzy and unclear. Therefore, the digitization
was mainly performed by key input. We digitized the AHMVs of the three components of declination (D), horizontal (H),
and vertical (Z) components. We designed a set of Excel templates to unify the data entry format. The digital templates are
very similar to the original data source to keep track of our work. The input templates include three workbooks, which are
used to store the AHMVs of one year, including the AHMVs of D component, H component and Z component. Every
AHMVs workbook consists of 14 worksheets, including text description, data worksheets from January to December of
every year and automatic summary worksheet. The monthly data worksheet header includes the station code, measuring
point ID, date, large value, 24 hourly mean values. The large value is a fixed value every month. The purpose of entering
a large value is to facilitate the rapid entry of each value. The 24 AHMVs can be calculated by adding the large values to
the 24 hourly mean values respectively. For example, the large value in January 1985 was 33,300. For each hourly mean
value of this month, we only need to input the digits after thousands' digit. If the input value of 0 hours on January 1 was
146.1, we can get the AHMV at this moment by 33300 plus 146.1, and so on. Missing values were marked as '99999'. An
example of the Excel tables with digitized data is presented in Fig. 2. The "key input" approach is slower but has the lower
error rate (Capozzi, 2020). After each month of data entry, we cross checked the digitized data with the original source
values in order to identify and remove transcription errors. Using this approach, it took us half a year to digitize the 1933 to
1954 data from paper records.

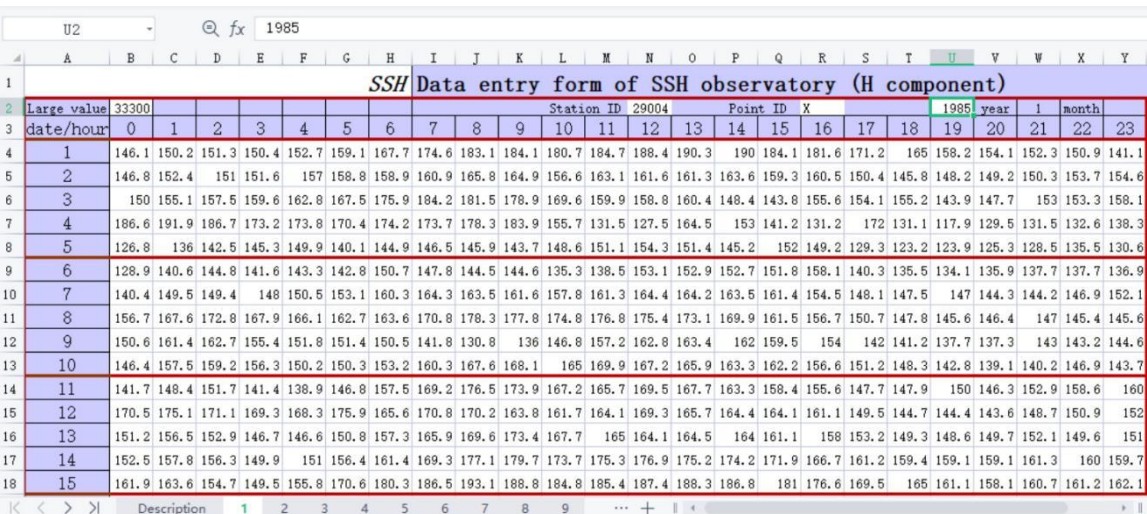

**Figure 2. An example of the Excel tables with digitized data**

## 2.3 Import data stored in various formats into Oracle Database

We developed a data convert software (Fig. 3) to import data stored in various formats (XLS, DBF, BAS and MDB) into a
unified Oracle database. In this way, all AHMVs were stored in the same database in a unified format. We call these stored
data without any correction as the original AHMVs data. It is convenient for the subsequent analysis and application. This
allows us to examine the data using Geomagnetic Data Processing Software (GDPST) developed by Geomagnetic Network
of China (zhang, 2016). GDPST was developed based on Oracle database. It provides a convenient way in data processing,
comparative analysis. The software also has the functions of the data query, data backup, and data download, etc.


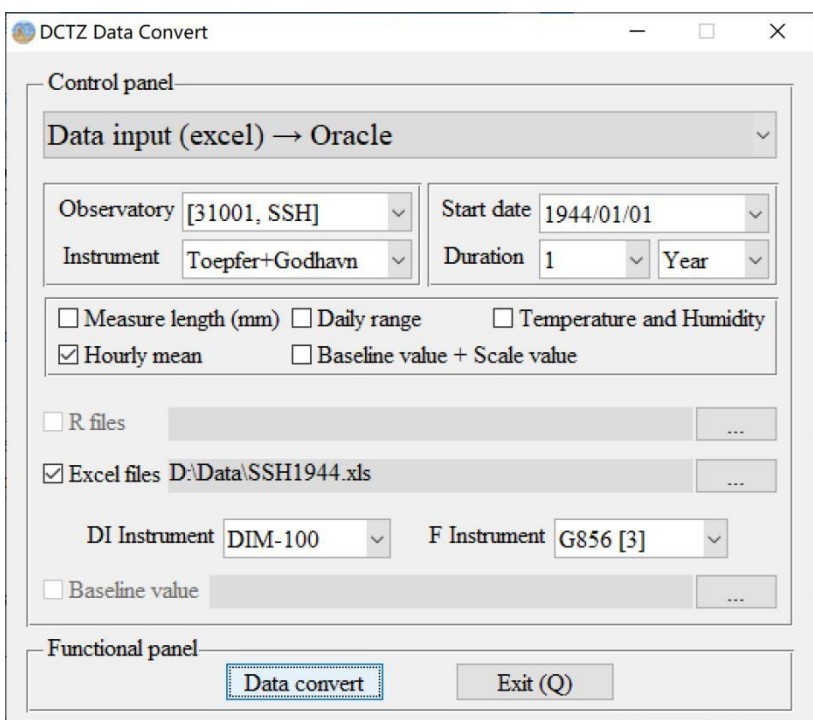

**Figure 3. Data import software**

## 3 Quality control of digitized data of SSH observatory

The purpose of quality control (QC) is to check the completeness and reliability of geomagnetic observation data. The quality of geomagnetic data is often affected by the changes in the instrument or environmental conditions of the measurements, for example repair or re-calibration of the instrument, instrument replacement, observatory relocation, gradual changes of the observation environment, changes in observing process, etc. (Morozova et al., 2014, Zhang et al., 2016). Most of such changes can lead to sudden breaks and jumps in the series of geomagnetic data, or gradual biases from the real geomagnetic characteristics. We call it data problems and define it as 'sudden breaks and jumps in the series of geomagnetic data, or gradual biases, or noise and change of transfer function etc.'. Correction of problem data before any subsequent analyses is highly desirable (Mestre, 2013).

In this study, the QC was performed in order to check the quality of the rescued data. The inspection contents include evaluating the completeness of data, the accuracy of daily variation, the stability of secular variation, and analyzing the influence factors of data quality.

## 3.1 the completeness of data

Based on the original AHMVs, the annual completeness is calculated, using the following formula (1):

$$C = (W_o - W_m)/W_o \qquad (1)$$

Here, $C$ is the completeness of the AHMVs, $W_o$ is the number of expected data in the chosen period, $W_m$ is the number of missing data, see Fig. 4 for the completeness of data. The series in this study consists of data measured from 1933 to 2019. It

has 5 larger gaps having a total of 66 months of data missing the number of missing data accounts for 6.5% of the total. All gaps were not replaced by interpolation in this study. War and instrument failure are the main reasons for the gaps of observation.

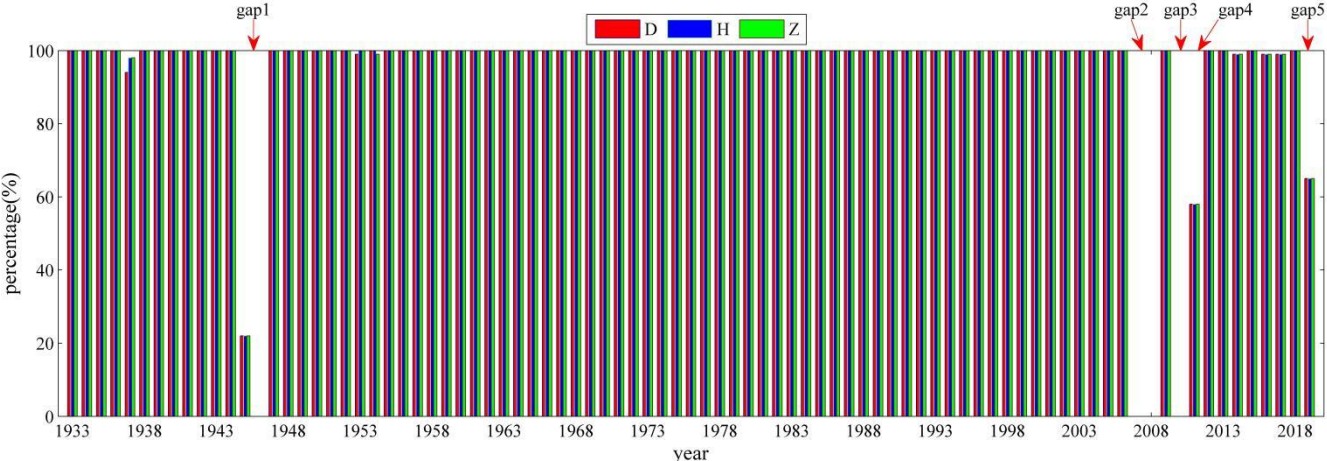

**Figure 4. The completeness of the AHMVs from 1933 to 2019**

**3.2 the accuracy of the rescued data**

We have designed a strict quality control procedure to ensure the accuracy of the rescued data. It consists of the following three steps:

1. Preliminary analysis of the series, detection of outliers.

In order to avoid the adverse impact of extreme data on the overall trend, we filtered out clearly obvious outliers by the

appropriate filtering function of Excel, such as the missing values which were marked as '99999', obvious input error, and so on.

2. Visual analysis of the series and their first-time derivative at different timescales.

After removing the obvious outliers, we plotted AHMVs of geomagnetic field components D, H and Z for all time from 1933 to 2019 (see Fig. 5). It can be seen from the figure that D, H and Z components have obvious trend. It is the secular

variation(SV)in geomagnetic field with time. The additional signal in the plots mainly comes from the activity of external field. The most significant influence is on both the horizontal components D and H; its influence on the vertical component is minor. In the plots, we do not see obvious step and peak interference.

We checked the AHMVs plots of SSH month by month and found that sometimes the geomagnetic changes were quiet and regular, sometimes violent and irregular, and most of the days the geomagnetic changes were superimposed on the regular quiet day changes with some disturbances of different shapes and amplitude. As shown in Fig. 6, taking the AHMVs record of January 1955 as an example, it can be seen from the figure that the record includes both regular periodic quiet day changes and complex perturbation, and the geomagnetic storms of January 18-19 are violent disturbances.

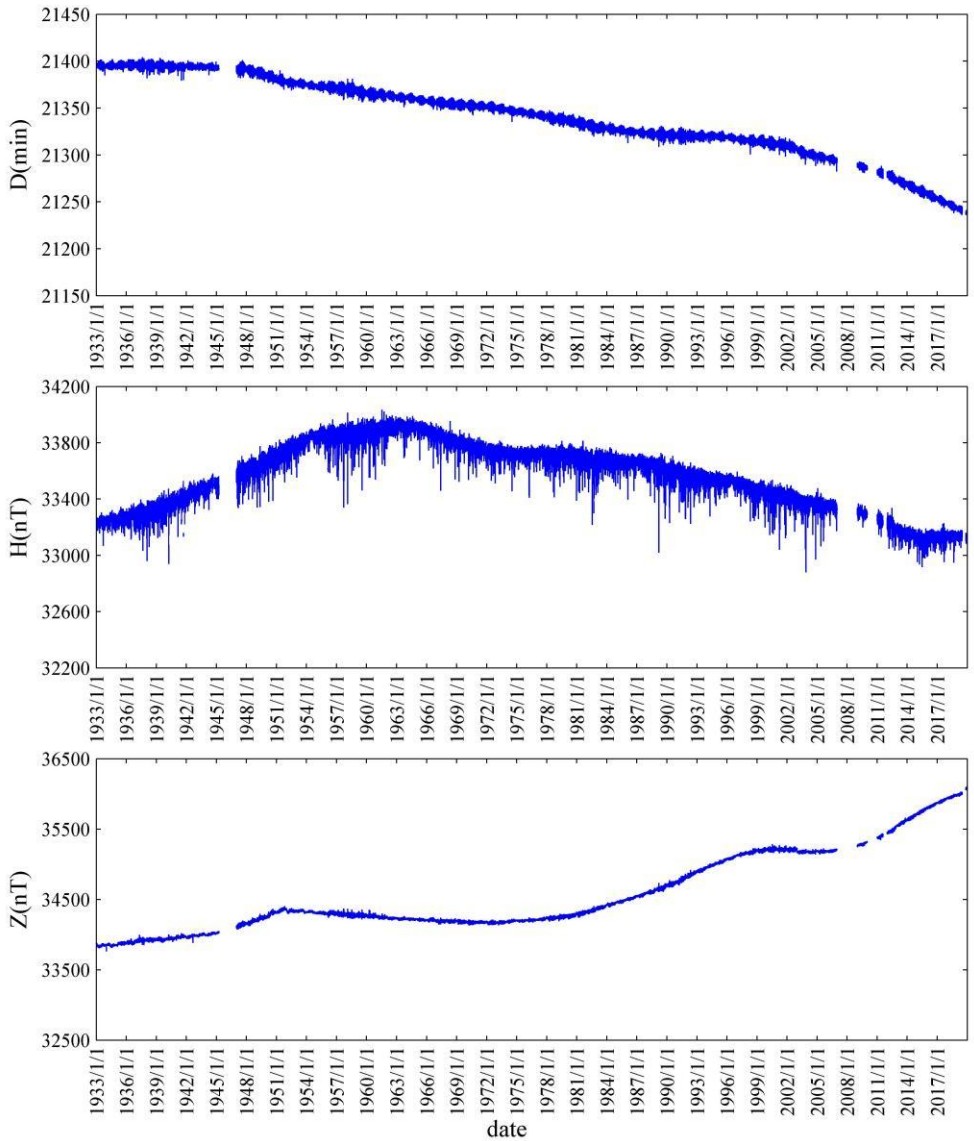

**Figure 5. The AHMVs plots of D, H and Z components for all time from 1933 to 2019**

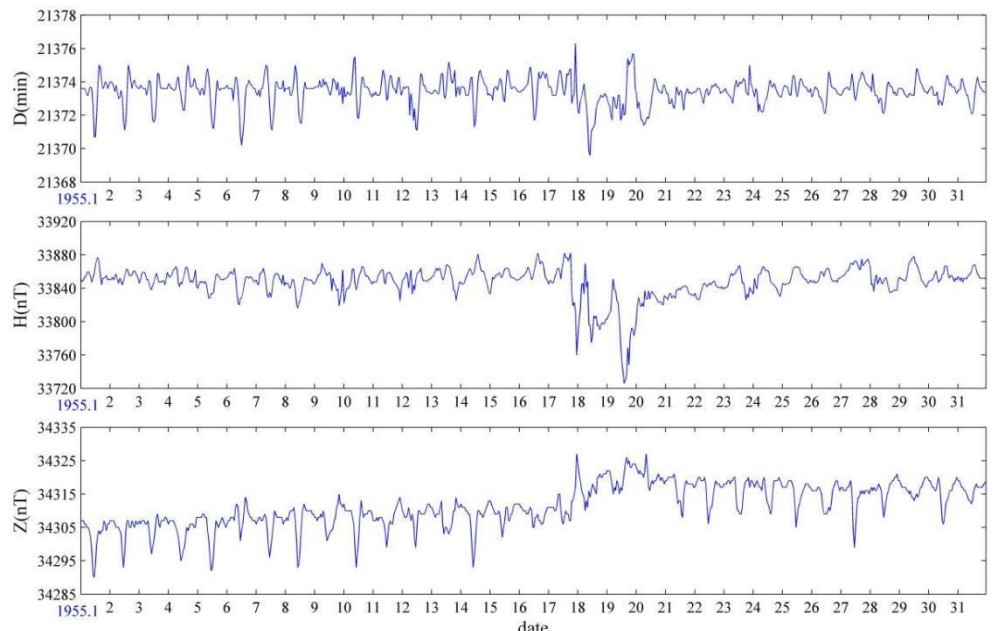

**Figure 6. The AHMVs record of January 1955 at SSH**

In order to eliminate the impact of trend and detect the data problems more effectively, we plotted the first-time derivative
(FTD) of AHMVs for D, H and Z components (Fig. 7). We calculated the FTD using the consecutive values of hourly series
(Morozova et al., 2014). For all geomagnetic components, the FTD is calculated as

$$dX/dt(\text{hour}) = (X(\text{hour}) - X(\text{hour} - 1))/1 \qquad (2)$$

Where $X$ is geomagnetic field components D, H and Z.

FTD plots are also particularly useful in evaluating artificial noise, especially interference in the shape of steps or spikes
(Linthe et al., 2013; Pang et al., 2013; Chen et al., 2014). It can be seen from the figure that the data after the FTD eliminates
the trend change, and the data are steady, going up and down within a certain range. ΔD varies between -13.4 min and 12.6
min, ΔH varies between -302 nT and 203 nT, and ΔZ varies between -70 nT and 62 nT.

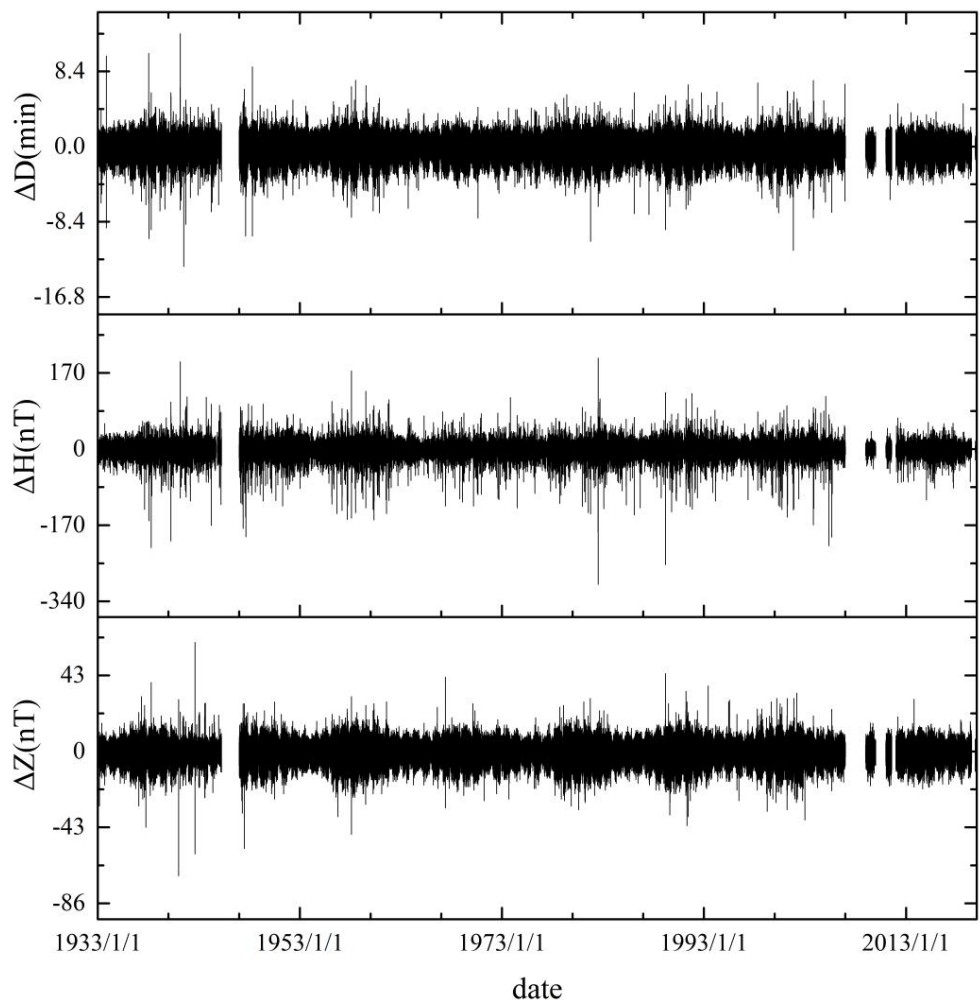

**Figure 7. The FTD plots of D, H and Z components**

200      3. The tolerance test detects the outliers and compare with geomagnetic indices.

For a large set of data with a normal or approximate normal distribution, 99.7% of the values are distributed in the ($\mu$-3$\sigma$, $\mu$+3$\sigma$) interval, where $\sigma$ and $\mu$ are the standard deviation and mean for all time. The values beyond this interval are generally considered as outliers. We presented the histograms of the FTD of D, H and Z components between 1933 and 2019 in Fig. 8, which aimed at detecting the outliers further. The distribution can be well modelled by the Gaussian probability density

205   function (red solid curve). The red vertical dashed lines indicate the lower and upper limits obtained by applying the criteria ($\mu$-3$\sigma$) and ($\mu$+3$\sigma$). We found that more than 98.6% of the FTD data points fall within the range of three times the standard deviations.

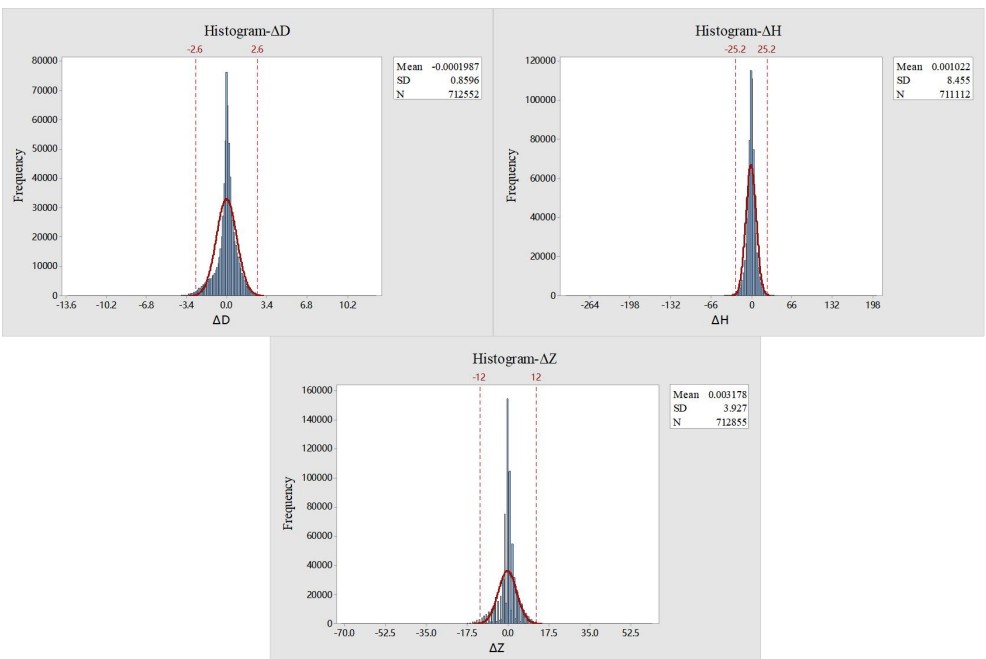

**Figure 8. The histogram of the FTD of between 1933 and 2019**

For the FTD data exceeding three times the standard deviation, we defined them as FTD outliers. We need to confirm whether the outlier is related to geomagnetic activity. Kp and Dst are two conventional indices to describe geomagnetic activity. Kp has no linear relationship with the geomagnetic activity, so the ap index was introduced. The ap is expressed in "ap units": 1 ap unit equals 2 nT (Menvielle et al., 2011). Kp and ap indices are produced currently by GeoForschung Zentrum (GFZ) Potsdam, Germany (Kp and ap values since 1932 are available on-line at https://www.gfz-potsdam.de/). Dst indices are produced currently by the World Data Center for Geomagnetism, Kyoto (Dst values since 1957 are available on-line at http://wdc. kugi.kyoto-u.ac.jp/). The comparative analysis (Fig. 9) also shows that the geomagnetic components, have a good correlation with Dst and ap indices, especially H and Dst. It should be noted that the H component in Figure 9 has eliminated periodic changes such as secular variation and seasonal variation.

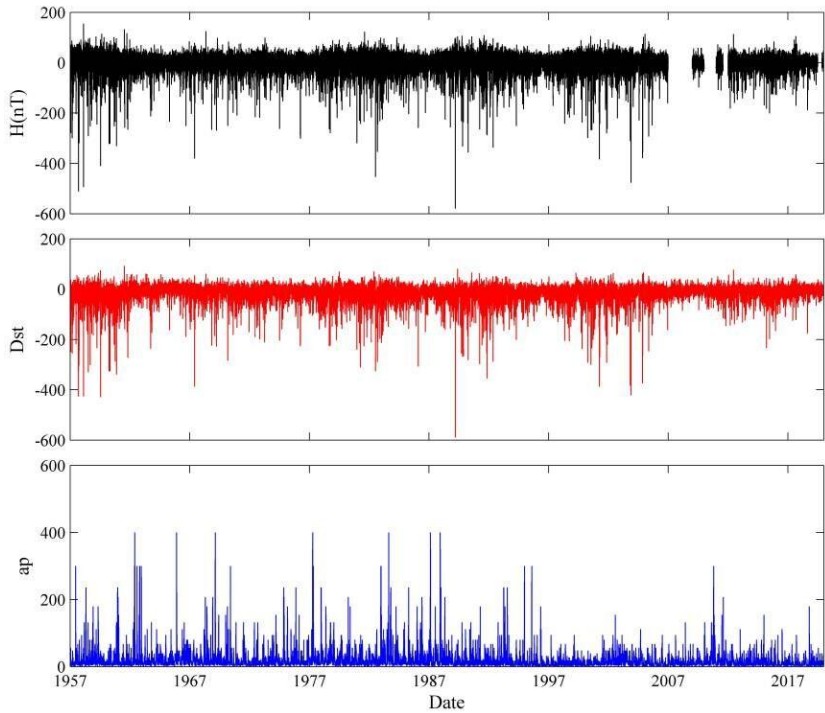

**Figure 9.** Comparative analysis of the H components with Dst and ap indices,

We compared the FTD outliers with the ap and Dst indices and tried to find out the cause of the FTD out of tolerance, and took corresponding measures: a) when the ap is greater than or equal to 24 nT, or Dst is less -30 nT, the outlier was considered to be caused by geomagnetic activity. The AHMVs at the corresponding time were not corrected. b) When the apis less than 24 nT and Dst is greater than -30 nT, we carefully looked for the cause of each FTD outlier by comparing the daily variation curves of multiple observatories (Sheshan, Chongming, Wuhan, Guangzhou, or Nanjing), and further consult the available documentation (*Observatory Communication Journal*, *Geomagnetic Observation Report*, *Chronicles of China Geomagnetic Observatory* and postal letters). Preliminary analysis found: for D component, 65.6% of the outliers were related to geomagnetic disturbance; the 33.5% showed no abnormality were found in the daily change curve; the remaining 82 values were questionable. For H component, 80.5% of the outliers were related to geomagnetic disturbance; the 17.6% showed no abnormality were found in the daily change curve; the remaining199 values were questionable. For Z component, 99.6% of the outliers were related to geomagnetic disturbance; the 0.4% showed no abnormality were found in the daily change curve; the remaining 112 values were questionable. A total of 393 FTD outliers were questionable and no relevant and useful information was recorded in the available documentation. The AHMVs at the corresponding time were not corrected but were marked as questionable data in the datasets, the quality flag was QC=Q. As shown in the Fig. 10, taking the FTD outliers of October 19, 2013 as an example, a clear deviation was found in the data between 8:00 to 13:00 from the real geomagnetic characteristics. Due to the lack of complete documentation, the questionable data were not corrected, just made the marks in the datasets, QC=Q. c) When the ap indices is less than 24 nT and Dst is greater than -30 nT, and a

change is registered in the available documentation. These data can be accepted to be corrected. In Table 2, we listed the date of the data to be corrected and the reasons recorded in the daily log and annual report. It took place only on 1 January 2003, a modern absolute instrument named Fluxgate Theodolite DIM-100 replaced Geomagnetic Induction Instrument Schulze, and an Overhauser Effect Proton Precession Magnetometer GSM-19F replaced Proton Precession Magnetometer CZM-2. These changes led to sudden steps in size of about 1.9', 8 nT, 40 nT and 35 nT in D, H, Z and F components respectively.

The log recorded the exact jump the measurements during the change of instruments at that time (SSH observatory, 2004).

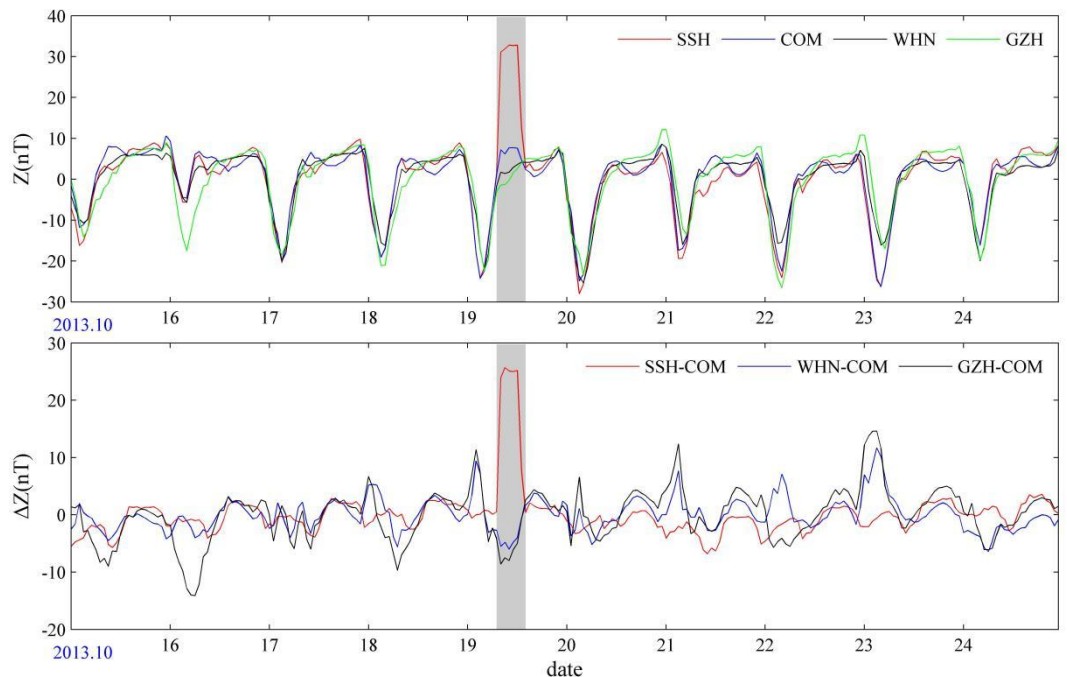

**Figure 10. The contrast curve of SSH, COM, WHN and GZH from** October **15 to 24, 2013**

**Table 2** Date and the reasons recorded of the data to be correction

| Date | Metadata | Time interval | Correction values (new values -old values) | | | |
|---|---|---|---|---|---|---|
| 2003.1.1 | Instrument replacements | 2003.1.1–2019.12.31 | D: 1.9' | H: 8 nT | Z: 40 nT | F: 35 nT |

### 4 Correction of the data problems

As was mentioned above, we corrected only the data of D, H, Z and F components that occurred after 1 January 2003. The break arose due to installation of new instruments in 2003.1.1. The corrected value is 1.9' for D component, 8 nT for H component, 40 nT for Z component and 35 nT for Z component (see table 2). As shown in Fig. 11 and Fig. 12, we gave the hourly and daily mean values curve of D, H and Z components before and after correction from January 1, 2001 to December 31, 2004. It can be seen that the quality of data has been greatly improved after correction.

The AHMVs curve shows obvious annual and seasonal variations. The seasonal variation shows the variation range is large in summer and small in winter. The ADMVs curve of D component shows an obvious long-term trend of slow decline from 2001 to 2004. No obvious change characteristics can be seen in Z curve.

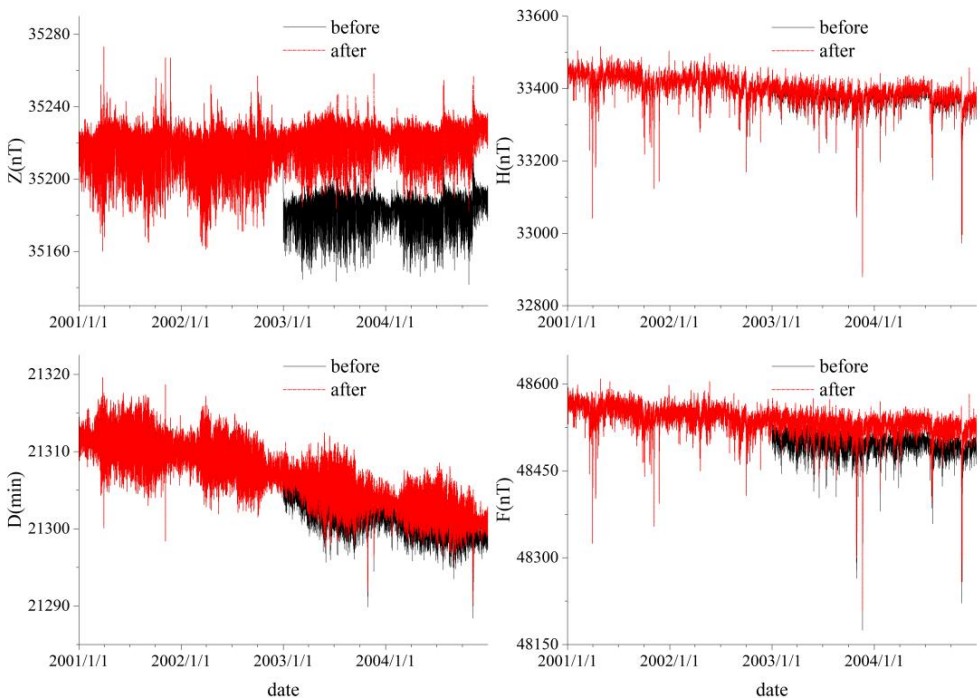

260                                 **Figure 11. Plot of hourly mean values before and after correction**

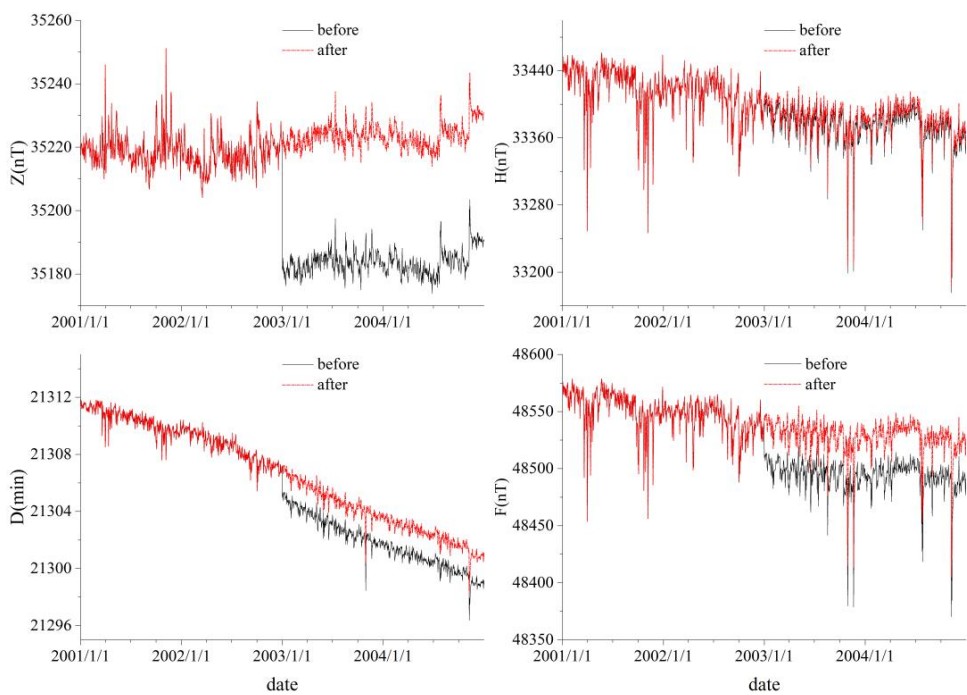

**Figure 12. Plot of daily mean values before and after correction**

Mutual comparison is an important method to check data quality. In general, the difference of the same component between two stations close to each other is small and stable. We also compared the data of SSH before and after correction with those data from COM which is the nearest observatory from SSH (Fig. 13). The differences of the three components before correction are: ΔD varies between -1.0 min and 2.4 min, ΔH varies between -2 nT and 14 nT, and ΔZ varies between -46 nT and 19 nT. The differences of the three components after correction are: ΔD varies between -0.3 min and 1.3 min, ΔH varies between -7 nT and 24 nT, and ΔZ varies between -20 nT and 4 nT. The standard deviations of the differences of the three components before correction are: 1.1 min, 3 nT and 20 nT. The standard deviations of the differences of the three components after correction are: 0.3 min, 3 nT and 3.3 nT. Again, it shows that the quality of observation data is improved after correction.

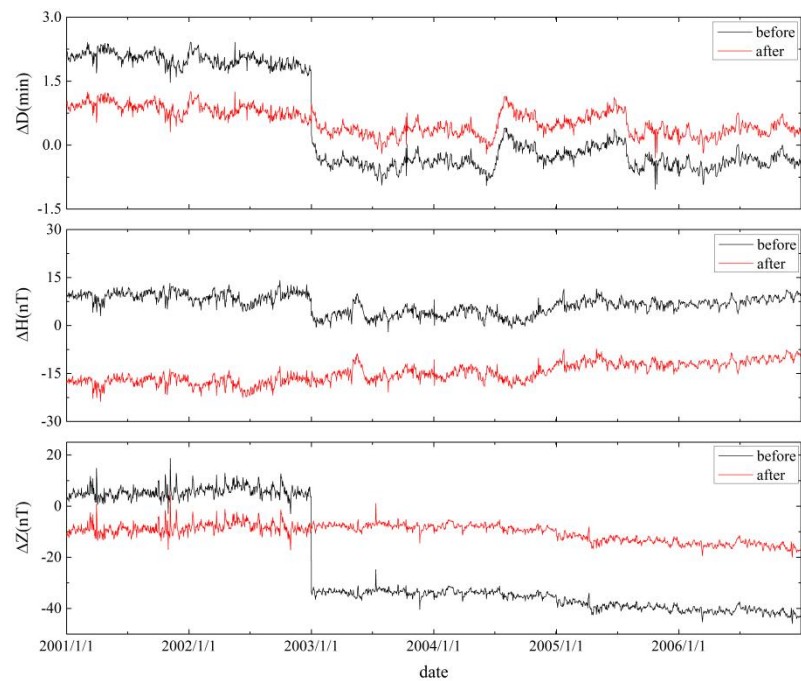

**Figure 13. Plot of difference between SSH and COM before and after correction**

## 5 Validation of the corrected series by compare with reference series

Inter-comparison of geomagnetic elements time series from adjacent observatories or geomagnetic models is also an important method to test accuracy and stability of data (Curto and Marsal, 2007). Firstly, we compared the rescued data with those data from the reference observatory. In China, the regular observation of most geomagnetic observatories began in the 1980s. Only eight geomagnetic observatories were established during the international geophysical year (Rasson, 2011). Among the eight observatories, Guangzhou observatory (GZH) is closer to Sheshan observatory. It started observation early,

and the quality of observation data were good. So, we selected GZH as the reference observatory. The GZH located in Guangzhou City, Guangdong Province, about 1240 kilometers northeast of SSH. It began geomagnetic observation in 1957. Due to the interference of Guangzhou metro operation, the new site Gaoyao Liantang Town was selected in 1996. The construction of the new observatory was completed at the end of 2001. The geomagnetic observation records officially began in new place on January 1, 2002.

We used GDPST which offers very useful diagnostic procedures of the data quality to plot inter-comparison of values curve and their difference curve from SSH and GZH observatories on hourly and daily timescales to detect data with potential quality issues year by year. As an example, we presented AHMVs, ADMVs and their difference curves of SSH and COM observatories in Fig. 14. At Fig. 14 in the upper panel (a) the AHMV and their difference curves are depicted, while in the lower panel (b) the ADMVs and their difference curves are plotted. On hourly scales the single components D, H and Z of

SSH and GZH behave roughly identical, and their difference series slowly fluctuates (due to geomagnetic activity) around a certain range. Spikes are caused in most cases by external disturbances. On daily scales the components D, H and Z show roughly identical, but their differences coincide clearly with the variation of the geomagnetic field. It is because the distance between two observatories too far to offset completely the influence of internal and external source fields in different regions.

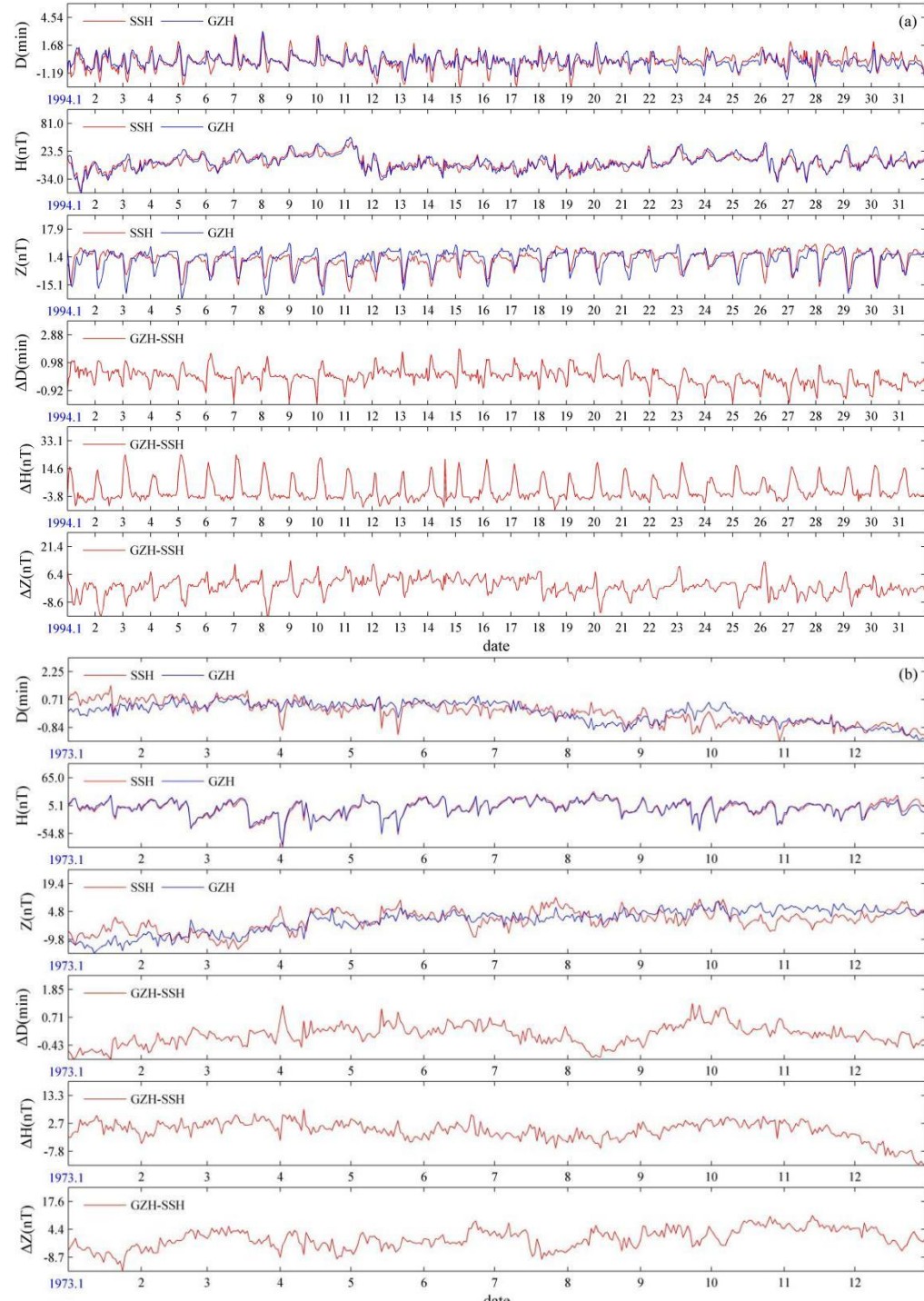


**Figure 14. AHMVs, ADMVs and their difference curves of SSH and GZH observatories. (a) the AHMVs and their difference curves. (b) the ADMVs and their difference curves**

Comparing the measured values with the calculated values of the model for a long-time scale is not only an important means to check the stability of secular variation, but also an important means to evaluate the accuracy of the model (Zhang, 2008b;

Thébault, 2010; Chen, 2012). One of the aims of geomagnetic observatories is the monitoring of SV (Reda et al., 2011). Secondly, to monitor the SV of SSH, we compared annual means values (AMV) series curve of X, Y, Z components calculated from the rescued records with these data calculated from the COV-OBS model (Gillet et al., 2013; Huder et al., 2020). The COV-OBS.x2 model covers the period from 1840 to 2020. The data source of the model is from observatory data, satellite data and the older surveys. The model can give the field contributions from the internal and external sources.

As can be seen from the Fig. 15, the change trends of X, Y and Z components from SSH and COV-OBS model are very consistent. X component increased year by year before 1962 and generally decreased after 1962; from 1933 to 2019, the Y component shows a general downward trend and the Z component shows a general upward trend. There are differences between the AMV of SSH and these data calculated from the COV-OBS model, the X component varies from -210 nT to -276 nT, the Y component varies from 17 nT to 94 nT, and the Z component varies from 198 nT to 289 nT. According to the

preliminary analysis, the main reasons for the large difference between SSH and COV-OBS model may be the local magnetic anomaly in Sheshan area, the uneven distribution of global stations, the lack of modeling data and data quality problems in SSH. This fully illustrates the importance of continuous and high-quality data in magnetic field modeling.

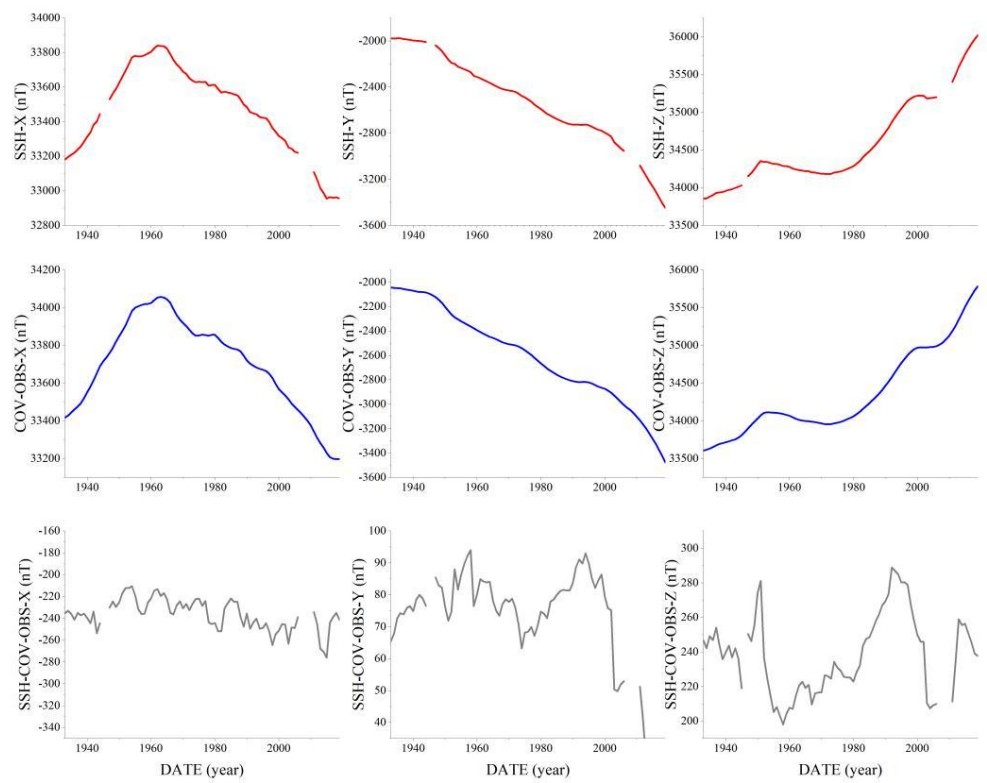

**Figure 15. The AMV and their difference curves of X, Y, Z components calculated from the rescued records and the COV-OBS model**

## 6 Application examples of SSH datasets

We calculated the first-time derivative using a difference between two consecutive years as SV and plotted the SV from the rescued data and COV-OBS model to detect possible geomagnetic jerks over the past 90 years. For all geomagnetic components, the SV is calculated as

$$\mathrm{d}X/\mathrm{d}t \text{ (year)} = (X \text{ (year)} - X \text{ (year-1)})/1 \tag{3}$$

Where $X$ is geomagnetic field components D, H and Z.

Geomagnetic jerks are defined as V-like or Λ-like changes in the SV and occur in a time period of a few months (Courtillot and LeMouël, 1984; Morozova et al., 2014; Kang et al., 2020). "The geomagnetic jerks are due to interactions of the core field and the rapid time-varying core flow" (Kuang et al., 2011). Since Malin and Hodder (1982), Courtillot and Le Mouël (1984) discovered the geomagnetic jerk in 1969, ten jerks have been detected in observatories from 1933 to 2020, of which 1969, 1978 (Alexandrescu et al., 1996), 1991 (De Michelis et al., 1998), 1999 (Mandea et a.l, 2000; Zhang et al., 2008a ), 2003 (Olsen et al., 2007; Feng et al., 2018; He et al, 2019), 2007 (Kotzé et al. 2010; Chulliat et al., 2010) and 2014 (Brown et al., 2016; Kloss and Finlay, 2019; Finlay et al., 2016; Kang et al., 2020) were global events. In addition, there were two local events, which occurred in 1949 (Mandea et al.,

2000) and 2011 (Chulliat and Maus, 2014; Kotzé and Korte, 2016). In 2017, there are similar characteristics of geomagnetic jerks, which may be a new geomagnetic jerk (He et al, 2019; Pavón- Carrasco et al., 2021). The jerks are more easily seen in the eastward component (Y) of geomagnetic secular variation. Not all jerks can be detected around the world, some seem to be seen only in limited regions (Morozova et al., 2014), and its occurrence time is not exactly same at each observatory.

Figure 16 gives the SV of the annual Y series from SSH and COV-OBS model, and the blue dotted line is the third-order moving average curve of Y component SV at SSH. Nine jerks are clearly seen in the plot. They occurred in 1950, 1971, 1978, 1993, 1999, 2004, 2008, 2013 and 2018. Except for the jerk occurred in 1978 and 1999, other events were a little later is seen at SSH. Between 2008 and 2017, only one jerk (in 2013) is seen at SSH, which is inconsistent with the study of other scholars mentioned above. They observed jerks event in 2011 and 2014 respectively. We cannot see a potential jerk in 2011 maybe because of the data gap until 2011.

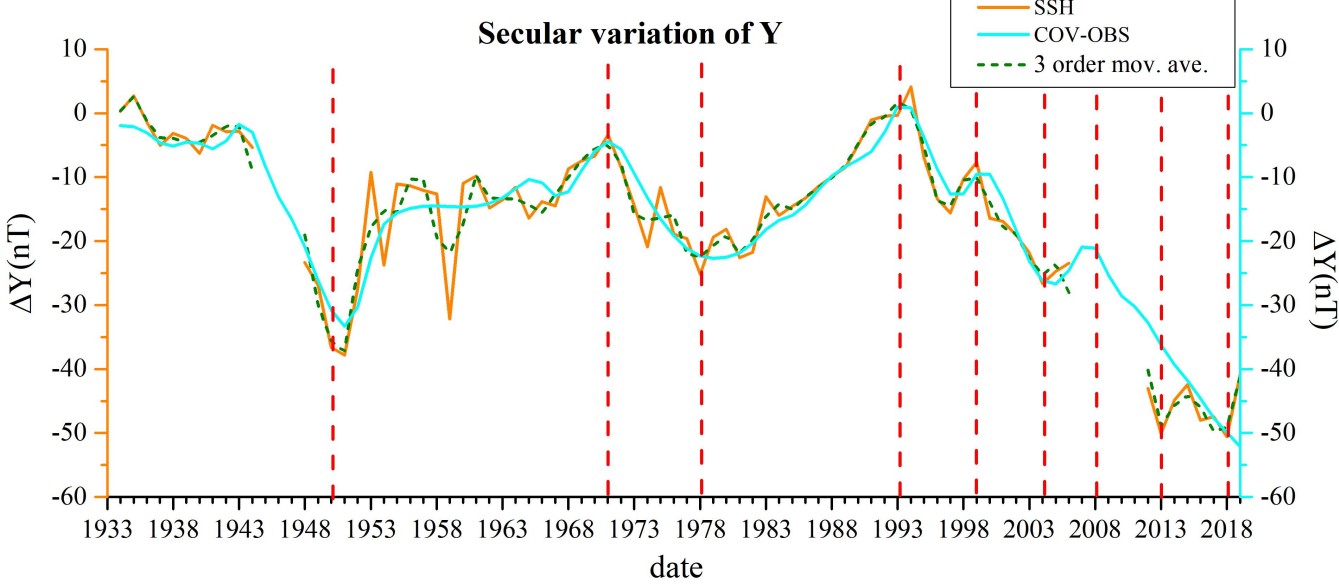

**Figure 16. The SV of the annual Y series from SSH and COV-OBS model**

A geomagnetic storm is a global phenomenon of magnetic disturbance. At low and mid latitudes, it mainly manifests itself as a decrease in the horizontal geomagnetic field (H) during a geomagnetic storm. According to the Kakioka Magnetic Observatory website, a total of 67 very large geomagnetic storms (variation range of H component > 300 nT) have occurred since 1933. Referring to the start and end time of the geomagnetic storm announced by the website, we studied the geomagnetic storms that recorded in the data series of SSH since 1933.We found a total of 42 very large geomagnetic storms (variation range of H component > 300 nT) in this data series (Table 3). Figure 17 shows an example of very large geomagnetic storm occurred on February 11, 1958. A storm sudden commencement (SSC) occurred at 1 a.m. on February 11, 1958. It is the sign of the start of the geomagnetic storm. This storm lasted about 53 hours and ended at 6:00 on the 13th. The maximum variation amplitude of D, H and Z during the geomagnetic storm is 210 nT, 649 nT and 133 nT respectively.

**Table 3** List of geomagnetic storms that have occurred at SSH since 1933

| SN | Start time | End time | variation amplitude | | | SN | Start time | End time | variation amplitude | | |
|---|---|---|---|---|---|---|---|---|---|---|---|
| | | | H | D | Z | | | | H | D | Z |
| 1 | 1937 08 22 03 | 1937 08 23 15 | 329 | 157 | 82 | 22 | 1978 08 27 02 | 1978 08 31 20 | 312 | 104 | 55 |
| 2 | 1938 01 22 02 | 1938 01 23 24 | 366 | 113 | 40 | 23 | 1982 07 13 16 | C | 538 | 156 | 56 |
| 3 | 1938 01 25 11 | 1938 01 27 10 | 333 | 160 | 39 | 24 | 1982 09 05 22 | 1982 09 08 02 | 385 | 148 | 68 |
| 4 | 1940 03 24 13 | 1940 03 26 08 | 534 | 109 | 76 | 25 | 1982 09 21 03 | 1982 09 23 21 | 306 | 100 | 61 |
| 5 | 1947 03 02 08 | 1947 03 04 22 | 338 | 74 | 68 | 26 | 1983 02 04 16 | 1983 02 06 20 | 303 | 88 | 33 |
| 6 | 1949 01 24 18 | C | 333 | 127 | 53 | 27 | 1986 02 06 13 | 1986 02 10 03 | 328 | 162 | 36 |
| 7 | 1949 05 12 06 | 1949 05 15 18 | 431 | 113 | 87 | 28 | 1989 03 13 01 | 1989 03 15 22 | 629 | 185 | 114 |
| 8 | 1950 03 19 05 | 1950 03 19 23 | 383 | 135 | 84 | 29 | 1989 10 20 09 | 1989 10 23 10 | 318 | 96 | 63 |
| 9 | 1957 09 13 00 | 1957 09 14 16 | 557 | 152 | 112 | 30 | 1990 04 09 08 | 1990 04 11 24 | 388 | 94 | 73 |
| 10 | 1957 09 29 00 | 1957 10 02 10 | 384 | 118 | 86 | 31 | 1991 11 08 06 | 1991 11 10 03 | 326 | 163 | 54 |
| 11 | 1958 02 11 01 | 1958 02 13 06 | 649 | 210 | 133 | 32 | 1992 05 09 19 | 1992 05 12 07 | 434 | 116 | 80 |
| 12 | 1958 07 08 07 | 1958 07 10 11 | 386 | 126 | 96 | 33 | 2000 04 06 16 | 2000 04 07 20 | 321 | 130 | 53 |
| 13 | 1958 09 03 08 | 1958 09 06 13 | 306 | 107 | 65 | 34 | 2000 07 15 14 | 2000 07 16 18 | 317 | 130 | 68 |
| 14 | 1959 07 15 08 | 1959 07 17 02 | 488 | 150 | 115 | 35 | 2001 03 31 00 | 2001 04 01 15 | 447 | 175 | 97 |
| 15 | 1960 03 31 09 | 1960 04 02 23 | 332 | 126 | 106 | 36 | 2001 11 24 05 | 2001 11 25 24 | 317 | 87 | 78 |
| 16 | 1960 04 30 12 | 1960 05 01 20 | 364 | 97 | 82 | 37 | 2003 10 29 06 | C | 330 | 160 | 50 |
| 17 | 1960 11 12 13 | 1960 11 14 23 | 387 | 140 | 49 | 38 | 2003 10 30 16 | 2003 11 02 21 | 323 | 140 | 42 |
| 18 | 1961 09 30 21 | 1961 10 01 20 | 340 | 47 | 33 | 39 | 2003 11 20 08 | 2003 11 21 24 | 461 | 100 | 39 |
| 19 | 1967 05 25 12 | 1967 05 29 20 | 459 | 116 | 80 | 40 | 2004 11 07 18 | C | 453 | 120 | 34 |
| 20 | 1969 03 23 18 | 1969 03 25 24 | 303 | 95 | 64 | 41 | 2004 11 09 18 | 2004 11 12 24 | 343 | 123 | 39 |
| 21 | 1976 03 25 13 | 1976 03 27 24 | 333 | 93 | 53 | 42 | 2005 05 15 02 | 2005 05 16 18 | 378 | 98 | 30 |
| note | C in "End time": followed by another storm | | | | | | | | | | |

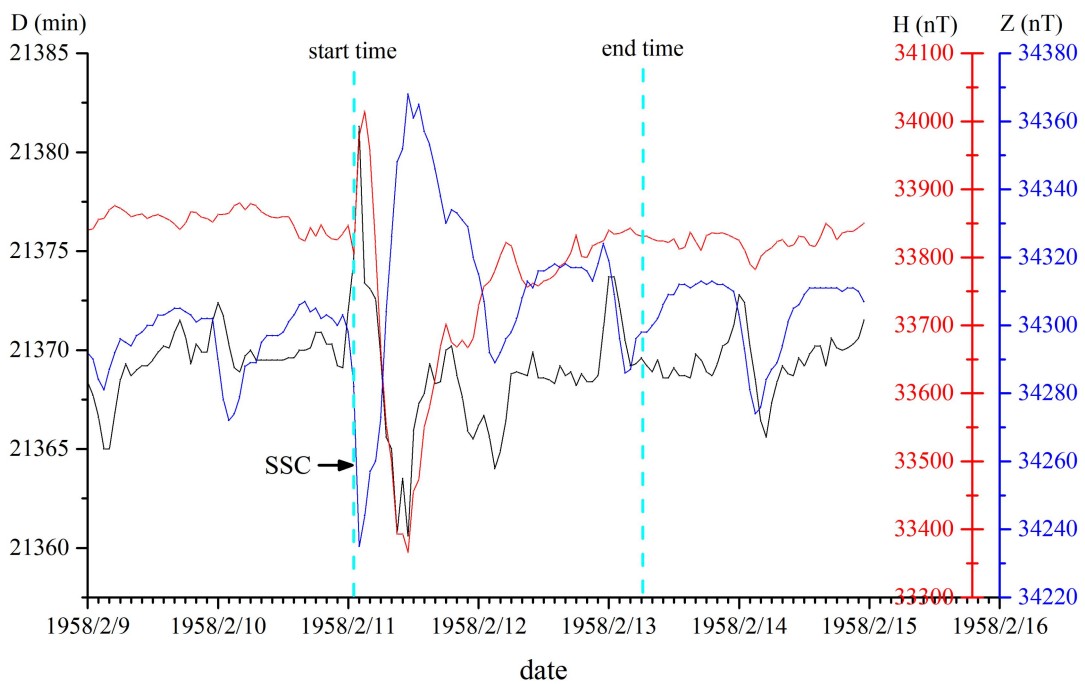

**Figure 17. The geomagnetic storm occurred on February 11, 1958**

## 7 Conclusion

This paper presents the acquisition process, the quality control, data correction, the quality examination and application examples of the datasets of SSH from 1933 to 2019. The quality examination results show that the corrected data have a good agreement with the reference observatory data and the model data. This fully indicates that the rescued data are of good quality. The datasets are valuable for studying the geomagnetic daily variation, geomagnetic field model construction and secular variation. It should be noted that the data marked with Q (QC=Q) are used with caution for the reasons mentioned above. A few problems were found in the acquisition of geomagnetic historical datasets: ① the rescue of paper data is a time-consuming and laborious work. For example, the font color of some reports is light, which makes automatic recognition difficult. We can only recognize and input by key. ② some metadata of the SSH has been missing, which brings difficulties to the identification and correction of data. Therefore, we believe that we should do our best to rescue historical data, to avoid the irreparable losses over time. Our plan for the next phase is to rescue the historical data from 1874-1932 at SSH, as well as from other observatories, to provide more high-quality data for the geomagnetic science community.

**Data availability**

The digitized and quality controlled AHMVs data are available at: https://doi.org/10.5281/zenodo.7005471 (zhang et al, 2022). The data are provided in Microsoft Excel format, including the observed absolute hourly mean values files of the three components (D, H and Z) at SSH for 1933-2019, and also the metadata files about the datasets.

**Author contribution**

SZ performed data correction, quality assessment and the analysis of application example. SZ prepared the manuscript. CF calculated X, Y, Z components from the COV-OBS model, drew pictures and performed revision of the manuscript. GZ was responsible for the digitization of paper data from 1933 to 1945 of SSH and provided digital series of the geomagnetic components 2009 and 2011. CC designed a set of Excel templates. JW developed the data import software. HS and GP collected and organized metadata. CG sorted out the geomagnetic storm list and geomagnetic indices.

**Funding information**

This work was funded by the National Key R&D Program of China (grant no. 2017YFC1500205), the National Natural Science of Foundation of China (grant no. 41974073) and the Macau Foundation and the pre-research project of Civil Aerospace Technologies of China (grant no. D020308).

**Competing interests**

The authors declare that they have no conflict of interest.

**Acknowledgement**

We thank SSH, GNC, and reference room of Institute of Geophysics, China Earthquake Administration for providing the valuable data resources. We are grateful to editors and anonymous reviewers for their helpful reviews. We sincerely thank all the staffs who have ever worked or are currently working at SSH. We also thank GeoForschung Zentrum Potsdam, Germany for providing online ap indices, World Data Center for Geomagnetism, Kyoto for providing online Dst indices, and the Kakioka Magnetic Observatory of the Japan Meteorological Agency for providing online magnetic storm catalog.

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
