# Peer review of "Rescue and quality control of historical geomagnetic measurement at SheShan Observatory, China"

_Earth System Science Data, 2022_

## Referee Comment (RC1)

This paper details the rescue and quality control of historical data at SSH observatory. As an application of the data set, the phenomenon of geomagnetic jerk is analyzed. This work has important reference significance for the processing and quality evaluation of observation data, and the rescued data is important in the study of geomagnetic field and its secular variation.

Here are some suggestions for revisions of the manuscript:

1. It is suggested to give the specific algorithms for some calculations in the manuscript, such as FTD.

2. The manuscript said that the homogeneity of corrected data has been greatly improved. Can you quantify it?

3. Using the COV-OBS model for comparative analysis, can you explain the reasons for choosing this model?

4. It is recommended that the Chinese characters on the pictures in the manuscript can be translated into English.

5. There are 5 gaps in Figure 4, it is recommended that these gaps should be marked in the figure.

---

## Author Response (AR1)

**Author's response to reviews on ESSD-2022-203**

The authors are thankful to the reviewers for their valuable comments. These comments are very helpful for revising and improving our manuscript. All the modifications are as follows. Corresponding changes have been made in the revised manuscript and are marked with "track changes".

**Referee #1:**

**General Comments:**

This paper details the rescue and quality control of historical data at SSH observatory. As an application of the data set, the phenomenon of geomagnetic jerk is analyzed. This work has important reference significance for the processing and quality evaluation of observation data, and the rescued data is important in the study of geomagnetic field and its secular variation.

**Thank you very much for your affirmation and valuable suggestions on our work.**

**Specific Comments:**

Here are some suggestions for revisions of the manuscript:

**Comment:** It is suggested to give the specific algorithms for some calculations in the manuscript, such as FTD.

**Response:** Thank you for your useful advice. We have given the specific algorithms for some calculations on now line 192 in the manuscript.

For all data series, the FTD is calculated as

$$dX/dt \text{ (hour)} = (X\text{(hour)} - X\text{(hour} - 1))/1$$

Where $X$ is geomagnetic field components D, H and Z.

**Comment:** The manuscript said that the homogeneity of corrected data has been greatly improved. Can you quantify it?

**Response:** Inter-comparison of geomagnetic elements time series from adjacent observatories is an important method to test accuracy and stability of data (Curto and Marsal, 2007). We added the comparison of the data before and after the correction of the SSH observatory and the COM observatory which is the nearest observatory from SSH on now lines 263-271. The differences of the three components before correction are: ΔD varies between -1.0 min and 2.4 min, ΔH varies between -2 nT and 14 nT, and ΔZ varies between -46 nT and 19 nT. The differences of the three components after correction are: ΔD varies between -0.3 min and 1.3 min, ΔH varies between -7 nT and 24 nT, and ΔZ varies between -20 nT and 4 nT. The standard deviations of the differences of the three components before correction are: 1.1 min, 3 nT and 20 nT. The standard deviations of the differences of the three components after correction are: 0.3 min, 3 nT and 3.3 nT.

Curto, J. J. and Marsal, S.: Quality control of Ebro magnetic observatory using

momentary values, Earth, Planets and Space, 59(11), 1187-1196, 2007.

**Comment:** Using the COV-OBS model for comparative analysis, can you explain the reasons for choosing this model?

**Response:** The geomagnetic field model COV-OBS.x2 covers the period 1840–2020. The data produce the model was from observatory series, satellite data, plus older surveys. The model can give the field contributions from the sources internal and external to Earth. We explained it on now lines 303-304 in the revised manuscript.

**Comment:** It is recommended that the Chinese characters on the pictures in the manuscript can be translated into English.

**Response:** We have translated all Chinese in figure 2 and figure 3 into English in the revised manuscript.

[Figure]

[Figure]

**Comment:** There are 5 gaps in Figure 4, it is recommended that these gaps should be marked in the figure.

**Response:** Thank for your advice. We have marked the gaps in the figure.

The gaps 1: 19450401-19461231; The gaps 2: 20070101-20081231; The gaps 3: 20100101-20101231; The gaps 4: 20110801-20111231; The gaps 5: 20190701-20191031.

[Figure]

**Referee #2:**

**General Comments:**

Suqing Zhang and coauthors have submitted a manuscript entitled 'Rescue and quality control of historical geomagnetic measurement at SheShan Observatory, China' to ESSD. The manuscript deals with the digitization of hourly mean values of SheShan Geomagnetic Observatory from 1933 to 2019.

**Comment:** The paper is a valuable contribution to the global geomagnetic observatory data set and the authors are to be commended for their efforts. The analysis presented in this manuscript is able to show some general data quality features and confirms to a certain degree that the data quality is good. The method of this analysis is sometimes of an ad hoc approach (like the outlier identification in Excel) and the authors would profit in future studies if they use more scientifically and geomagnetically motivated methods like comparison of the H component with the Dst index and the use of the time derivatives of the hourly means. I recommend minor revisions for the present manuscript as indicated below.

**Response:** Thank you very much for your affirmation and valuable suggestions on our work. According to your suggestions, we added the comparison of the H component with the Dst index and the use of the time derivative of the hourly mean in the revised manuscript.

**Specific Comments:**

**Comment:** The abstract is well written and the English reads very well. Still, for publication its language needs improvement. Even in the abstract, there are a number of mistakes including missing and superfluous blanks. I will not mark or correct language mistakes in detail here, as they are too many. But before publication, it is the

duty of the authors to read again the complete article and correct the language. I am happy to read the article once more after that.

**Response:** Thank you for your comment and careful review. We carefully corrected the language mistakes in the revised manuscript.

**Comment:** I have not heard about zenedo before, but it looks like it is supported by Cern and is a trustworthy data archive and distribution instance.
You provide, through zenedo, a 50 MB excel spreadsheet. My Office package crashed when I tried to load this file. My estimate is that if you provide HMVs (hourly mean values) in a simple ASCII format, then 100 years of data will be only about 10 MB in file file size. Excel seems to be not an ideal format. In any case, smaller yearly files would be more appropriate.

**Response:** According to your suggestions, we split the 50 MB excel file into 9 small files. Data for every 10 years is as a file. We also updated the datasets at zenedo. The Updated dataset are available at: https://doi.org/10.5281/zenodo.7005471

**Comment:** Line 56
What is the meaning of 'correction of the selected homogeneity'?

**Response:** It means "correction of problem data with clear reasons recorded in the documents". Corresponding changes have been made on now line 58 in the revised manuscript and are marked with "track changes".

**Comment:** Line 64
What is 'reference room'?

**Response:** The reference room is a resource center, used to collect books, journals, papers, monographs, Unpublished reports, internal textbooks, research reports, reference documents and scientific research achievements related to the discipline. We explained it on now lines 67-69 in the revised manuscript.

**Comment:** line 69
What is the DBF format?
Same for BAS and MDB, please explain in the text.

**Response:** DBF, BAS and MDB are all data file storage formats. The DBF is a tabular data file stored in binary and is the database format used by dBase and FoxPro databases in DOS systems. The BAS file format is written in the BASIC language, a plain-text data storage format. The MDB format is a storage format used by Microsoft Access software that can generally be opened directly with ACCESS. We explained them on now lines 75-78 in the revised manuscript.

**Comment:** line 102
Please rewrite 'useful for re-verifying the data when the documents were not available', it took me a long time to understand what you want to say. Also, I guess that for old paper copies it is not good to be carried around too much and as soon as you have a digital picture, which is fast to make, you can bring the respective paper again to its normal archive place with the usual temperature, humidity etc.
Also, which character recognition program did you use?
I agree that typing the numbers by humans is often a good solution.

**Response:** Yes, your understanding is correct. We rewrote this sentence on now lines 110-113 in the revised manuscript.

The character recognition program we used is ABBYY.

**Comment:** line 111

What is 'large value'?

I think what you mean is a 'base value', to be added to the tabulated values to get the AHMVs. You need to explain what the base value is as most people do not know the concept.

**Response:** The 'large value' is similar to the 'base value', but not the 'base value'. This value is a fixed value every month. The purpose of entering a large value is to facilitate the rapid entry of each hourly mean value. We explained it on now lines 122-126 in the revised manuscript.

**Comment:** line 115

I do not understand:

'After half a year's efforts, we input the data from 1933 to 1954 into the Excel template and completed the digitization of paper data.'

Do you mean:

'Using this approach, it took us half a year to digitize the 1933 to 1954 data from paper records.

**Response:** Yes, your understanding is correct. We mean: 'Using this approach, it took us half a year to digitize the 1933 to 1954 data from paper records.' We modified it on now lines 129-130 in the revised manuscript.

**Comment:** line 125

I would move the reference to Zhang 2016 behind 'Geomagnetic Network of China', to give the reader a reference for the Geomagnetic Network of China.

**Response:** corrected.

**Comment:** line 137

I agree that a very important property of geomagnetic observatory data (and geophysical time series in general) is to be homogeneous. But I disagree with the use of the term 'inhomogeneities' in this context, even though it was used by Morozova et al (2014), too. For example, if you have a time series of 10 years and the first 5 years it was recorded by an instrument A and the last 5 years by an instrument B and their transfer functions are different. I agree that the time series is not homogeneous, but I do not know what would be the homogeneous part of the time series and what part is the inhomogeneity.

Suggestion: call it data problems and define it as 'sudden breaks and jumps in the series of geomagnetic data, or gradual biases, or noise and change of transfer function etc.'

**Response:** We agree with you and make corresponding amendments on now lines 149-150 in the revised manuscript.

**Comment:** line 142

continuity -> completeness, or data availability

**Response:** corrected. continuity -> completeness

**Comment:** line 147
having a total of 67 monthly data missing -> having a total of 67 months of data missing
**Response:** corrected.

**Comment:** line 148
Remove '. "Every trial to correct data can produce unwished secondary effects in the result" (Linthe, 2013),' if you do not have data, you cannot correct data. Data interpolation is not permissible for AHMVs as the temporal change in the magnetic field is happening on periods longer than 1 hour and shorter than the length of your data gaps.
**Response:** removed

**Comment:** line 163
The noise -> The additional signal
**Response:** corrected.

**Comment:** line 209
AP -> Ap
**Response:** corrected.

**Comment:** page 13
The FTD of AHMVs should have been analyzed instead of FTD of ADMVs, as the daily means are much more smoothed than the hourly means.
**Response:** Thank you again for your valuable advice. On now lines 189-245 we analyzed the FTD of AHMVs instead of FTD of ADMV and more details have indeed been found.

**Comment:** line 247
corrected -> correction
**Response:** corrected.

**Comment:** page 14
A jump in the data cannot be corrected by determining the difference of the mean 3 month prior and 3 months after the jump. With this approach, you include errors from secular variation, from seasonal variation and from geomagnetic disturbance. You are not removing the jump-problem, you just make it less visible by subtracting a value close to the unknown jump. I recommend to not correct the data here, or to determine the exact jump from the measurements during the change of instruments
**Response:** Your suggestion is very helpful. The exact jump from the measurements during the change of instruments. With the efforts of the staff of SSH observatory, the daily log of 2004 was finally found. The log records the exact jump the measurements during the change of instruments at that time. We explained in the revised manuscript and corrected the data published at zenedo accordingly.
**Comment:** line 264

Jump outliers are best detected by time derivative of AHMVs, not by comparison with models.
**Response:** Yes. We deleted the sentence in the revised manuscript.

**Comment:** line 299
and so on -> and data quality problems in SSH.
**Response:** corrected.

**Comment:** line 322
You can not see a potential jerk in 2011 because of your data gap until 2011.
**Response:** Yes, you are right. We explained it on now lines 337-338 in the revised manuscript.

**Comment:** line 330
You can also include the study of geomagnetic storms.
**Response:** Thank you for your useful advice. We added the study of geomagnetic storms on now lines 341-354 in the revised manuscript.